# A Holistic Evolutionary and 3D Pharmacophore Modelling Study Provides Insights into the Metabolism, Function, and Substrate Selectivity of the Human Monocarboxylate Transporter 4 (hMCT4)

**DOI:** 10.3390/ijms22062918

**Published:** 2021-03-13

**Authors:** Eleni Papakonstantinou, Dimitrios Vlachakis, Trias Thireou, Panayiotis G. Vlachoyiannopoulos, Elias Eliopoulos

**Affiliations:** 1Laboratory of Genetics, Department of Biotechnology, School of Applied Biology and Biotechnology, Agricultural University of Athens, 75 Iera Odos, 11855 Athens, Greece; eleni.ppk@gmail.com (E.P.); dimvl@aua.gr (D.V.); thiraiou@aua.gr (T.T.); 2University Research Institute of Maternal and Child Health & Precision Medicine, and UNESCO Chair on Adolescent Health Care, National and Kapodistrian University of Athens, Aghia Sophia Children’s Hospital, 8 Levadias Street, 11527 Athens, Greece; 3Lab of Molecular Endocrinology, Center of Clinical, Experimental Surgery and Translational Research, Biomedical Research Foundation of the Academy of Athens, 11527 Athens, Greece; 4Department of Informatics, Faculty of Natural and Mathematical Sciences, King’s College London, Strand, London WC2R 2LS, UK; 5Department of Pathophysiology, Medical School, National and Kapodistrian University of Athens, 15772 Athens, Greece; pvlah@med.uoa.gr

**Keywords:** cancer metabolism, transmembrane proteins, monocarboxylate transporter 4, evolutionary study, 3D molecular modelling, pharmacophore design

## Abstract

Monocarboxylate transporters (MCTs) are of great research interest for their role in cancer cell metabolism and their potential ability to transport pharmacologically relevant compounds across the membrane. Each member of the MCT family could potentially provide novel therapeutic approaches to various diseases. The major differences among MCTs are related to each of their specific metabolic roles, their relative substrate and inhibitor affinities, the regulation of their expression, their intracellular localization, and their tissue distribution. MCT4 is the main mediator for the efflux of L-lactate produced in the cell. Thus, MCT4 maintains the glycolytic phenotype of the cancer cell by supplying the molecular resources for tumor cell proliferation and promotes the acidification of the extracellular microenvironment from the co-transport of protons. A promising therapeutic strategy in anti-cancer drug design is the selective inhibition of MCT4 for the glycolytic suppression of solid tumors. A small number of studies indicate molecules for dual inhibition of MCT1 and MCT4; however, no selective inhibitor with high-affinity for MCT4 has been identified. In this study, we attempt to approach the structural characteristics of MCT4 through an in silico pipeline for molecular modelling and pharmacophore elucidation towards the identification of specific inhibitors as a novel anti-cancer strategy.

## 1. Introduction

Cancer is a multifactorial disease defined by uncontrolled cell growth and metastasis [1]. Some of the factors that have been associated with the pathogenesis of cancer include the activation of oncogenes, the inactivation of tumor-suppressing genes, epigenetic modifications, and mutagenesis caused by external factors. The complexity of cancer creates numerous hurdles in drug development, such as off-target side effects, selective distribution to cancerous tissues, multidrug resistance, and cell permeability of drugs [2]. The development of new anti-cancer drugs should follow a multidisciplinary approach in order to provide pharmacological molecules that are both safe and efficient, and advanced in silico drug design approaches are being employed in modern research to predict and/or enhance the effectiveness of potent therapeutic inhibitors and simulate the level of interaction of proposed molecules with a specific anti-cancer drug target [3].

One of the hallmarks of cancer is the reprogramming of cellular metabolism and the switch from oxidative phosphorylation to glycolysis for the required energy and nutrients during tumorigenesis and metastasis [4]. Tumors are characterized by distinct metabolic activities that depend on their access to nutrients, their biological actions, and their spatiotemporal localization [5]. The majority of body cells is oxidative and fully convert glucose to carbon dioxide, while cells exposed to hypoxia and proliferating cells mostly convert glucose to lactate through anaerobic and aerobic glycolysis, respectively. Researching these metabolic differences is vital in cancer biology. In solid tumors, the processes associated with adaptation to hypoxia and cell proliferation function through different mechanisms. Hypoxic adaptation involves hypoxia-inducible factors (HIFs) and is a survival mechanism, while metabolic adaptation to cell proliferation involves growth factors and their respective effectors [6]. Monocarboxylates are a fundamental energy source of animal cells with their ability to be produced without oxygen consumption [7]. 

The transfer and handling of carboxylates are facilitated by proton-linked monocarboxylate transporters (MCTs), which are encoded by the solute carrier gene family *SLC16A* [8]. MCTs promote the metabolic cooperation between different tissues and are essential in sustaining energy balance and pH homeostasis. Specifically, MCT-mediated lactate transportation into the cell (influx) and out of the cell (efflux) is of high metabolic importance. Lactate efflux is essential for cells and tissues, which generate substantial quantities of lactic acid as a result of glycolysis. At the same time, lactate influx is vital in tissues where lactate is being used as a respiratory fuel, a gluconeogenic substrate, or a glycerol–neogenic substrate. Lactate exchange between adjacent cells is termed “lactate shuttle”.

MCTs are of great interest in anti-cancer research due to their main role in lactate metabolism [5]. They are overexpressed in cancer cells and are identified for their multiple activities such as metabolic signaling and metastasis, as well as biomarkers in certain cancer types [9]. Monocarboxylate transporter 1 (MCT1) and monocarboxylate transporter 4 (MCT4) are the most widely expressed isoforms in cancer cells [10]. Particularly, MCT4 is expressed among different cancer cells, such as breast, colon, prostate, and gliomas. MCT4 exerts a variety of activities in cancer, including metabolic exchanges, metabolic signaling, and cancer metastasis. The leading facilitator of MCT4s’ influence on cancer is lactate. As has been already mentioned, MCT4-expressing cells produce large amounts of lactate [11]. Lactate is a prominent fuel for oxygenated cancer cells [12]. In a hypothesis termed “metabolic symbiosis”, lactate produced by glycolytic MCT4-expressing cells is used by oxidative MCT1-expressing cells that use lactate in mitochondrial metabolism. This supposed mechanism gives oxidative cancer cells various advances, such as high production of ATP, compared to aerobic glycolysis.

MCT4 is mainly expressed in tissues that rely on glycolysis, such as white skeletal muscle fibers, astrocytes, and white blood cells. MCT4- expressing cells, both healthy and pathological, are the most substantial lactate producers [11]. Although MCT4 is primarily located at the plasma membrane, it is not glycosylated post-translationally. Instead, MCT4 requires an ancillary protein, specifically basigin, for its correct translocation to the plasma membrane. There, the transporter’s key function is exporting lactic acid derived from glycolysis. MCT4 showcases a lower affinity than MCT1 for the majority of substrates and inhibitors [13]. Notably, its high affinity for pyruvate could be significant, as it restricts the loss of pyruvate from the cell, which, were it to happen, would block the removal of nicotinamide adenine dinucleotide (NADH) that is produced in glycolysis through reduction of pyruvate to lactate. Furthermore, its low affinity for lactic acid restricts its efflux from skeletal muscles as exercise levels increases and drops the muscle pH [14]. In excessive lactic acid production, the pH drops significantly, and fatigue occurs. This mechanism prevents further lactic acid production, which could have adverse effects on the organism by causing systemic lactic acidosis. Concerning the regulation of its expression, MCT4 seems to be highly influenced by conditions such as hypoxia. It appears that hypoxia-inducible factor 1alpha (HIF-1alpha) enhances MCT4 gene transcription in response to hypoxia and upregulates the transporter’s expression [15]. The aforementioned attributes that MCT4 showcases appear to play an important role in cancer.

Several potential inhibitors of MCTs have been reported; however, they do not have an absolute specificity for their various isoform MCTs. The first inhibitors to be detected and recorded were phloretin, flavonoids such as quercetin, stilbene disulphonates, and α-cyano-4-hydroxycinnamate (CHC) and its analogues. CHC has been the focus of a large number of in vitro and in vivo studies showing its therapeutic effects. However, two of its disadvantages are its decreased specificity for a given MCT isoform and its ability to inhibit the mitochondrial pyruvate and anion exchanger 1, as well as the fact that it is usually active in the upper micromolar range [16]. In more recent studies, other inhibitors with higher affinity for MCTs have been reported. More specifically, AR-C155858, a cell-permeable thieno[2,3-d]pyrimidinedione compound, has been developed that acts as a potent inhibitor of MCT1 and monocarboxylate transporter 2 (MCT2), as well as AZD3965, a pyrrole pyrimidine derivative that represents a second-generation MCT1/2 inhibitor, currently in clinical trials of Phase I/II for advanced solid tumors in the prostate and stomach and diffuse B-cell lymphomas [17,18]. Despite this, both inhibitors have limited therapeutic effect, as there is the compensatory effect of MCT4, which they do not inhibit [19]. In their study, Draoui et al. identified another family of potential inhibitors, 7-aminocarboxycoumarine, which strongly inhibits MCTs. From this family, the main compound 7-(*N*-benzyl-*N*-methylamino)-2-oxo-2H-chromene-3-carboxylic acid (7ACC2) appears to selectively inhibit only lactate uptake but not its export from cancer cells that express both MCT1 and MCT4 [20]. The most prominent inhibitor reported to date is an anti-hypertensive drug, syrosingopine, that is found to inhibit simultaneously MCT1 and MCT4, preventing lactate efflux [21]. Nonetheless, there is a great need to find special inhibitors of MCTs, as their role in the development of the tumor is crucial due to their important activities and their overexpression in cancers compared to healthy tissues [10].

In this study, we attempt to elucidate the structural features of MCT4, for specified substrate recognition, through an in silico molecular modelling pipeline as a therapeutic approach towards the identification of potent inhibitors of MCT4 as a significant protein in cancer recognition and metabolic pathways.

## 2. Results

### 2.1. Evolutionary Study and MCT4 Specific Conserved Motifs Elucidation

Multiple sequence alignment of the *SLC16* family members from mammals was used initially for the construction of the neighbor-joining (NJ) phylogenetic tree of MCT4 across species (Figure 1 and Appendix A) and to identify highly conserved regions and sequence motifs that have been reported for the monocarboxylate transporters and the MFS superfamily (Figure 2). A more focused evolutionary study of the mammalian MCT4 is shown in Appendix A. It was concluded that the *N*-terminal half of the protein (TM1–TM6) shows a higher level of conservation, whilst the C-terminal half (TM7–TM12) denotes a lower percentage of conserved residues. The two six-helix halves of the molecule have different functionality, with the *N*-terminal domain participating mainly in proton binding and membrane insertion, and the C-terminal domain differentiates in each isoform for specific substrate binding. Across the major facilitator superfamily, motif [RK]XGR[RK] (referred also as Motif A), repeated in loops L2-3 and L8–9, is evident in almost all subfamilies [22]. This sequence motif in loop L2-3 can be traced as XXGR[RK] in the SLC16 family, whereas in loop L8-9 it cannot be identified. However, the hMCT4 sequence of the corresponding region is evident as 287-LGKVR-291 with Arg-291 being fully conserved across MCTs (Figure 2). Three sequence signatures have been reported for the *SLC16A* family. At the *N*-terminal and at the beginning of cytosolic TM1, Motif 1 [D/E]G[G/S][W/F][G/A]W is fully conserved within the family, can be extended with a preceding proline residue, and is implicated in a structural rearrangement during proton binding. A conserved sequence at the start of TM5 YFXK[R/K][R/L] XLAX[G/A]XAXAG also stands out and is denoted as Motif 2 in Figure 2, as well as Motif 3 LXGPPXXGXLXD, which is fully conserved and located on the proposed exofacial boundary of TM11.

### 2.2. Three-Dimensional Homology Modelling of MCT4

The three-dimensional model of MCT4 was based on the *Homo Sapiens* glucose transporter GLUT1 (PDB: 4PYP) and the *Syntrophobacter fumaroxidans* MPOB SLC16 homolog (PDB: 6G9X) (Figure 3a) [23,24]. The motifs of the *SLC16A* and MFS family were recognized in the expected region of the model (Figure 3b). The all-atom RMSD of the hMCT4 model and hGLUT1 was 2.17 Å, and the all-atom RMSD of the hMCT4 model and SfMOB was 2.85 Å. We validated the geometry and 3D conformation of the obtained model observing 392 (96.6%) residues in the favored region, 11 (2.7%) residues in the allowed region, and 1 (0.2%) residue in the outlier region, whilst the planarity, omega torsion, rotamers conformation, and atom clashes reported were less than 0.5%. The resulting model was in an outward-open conformation, and the key residues reported in the literature to participate in proton interaction and substrate translocation were found to be exposed in the canal. In particular, Lys40, Phe130, Tyr332, Asp274, Arg278, Leu360, and Glu363 are involved in proton binding, and Phe248, Lys254, Asp255, Tyr384, and Asp375 mediate the substrate binding (Figure 3c). The model was then embedded in a lipid bilayer, and the system was solvated and subjected to energy minimizations and molecular dynamics simulations (Figure 3d). The all-atom RMSD of the hMCT4 model with the very recently published crystallographic structure of human monocarboxylate transporter 1 (MCT1) (PDB: 7CKR) is 0.936 Å (Appendix A).

### 2.3. MCT4 Specific Multimodal Pharmacophore Design

A 3D pharmacophore model was generated for the substrate channel of MCT4 as a molecular model that ensembles all steric and electronic features that are necessary to ensure optimal interaction. The pharmacophoric features included positively or negatively ionized regions, hydrogen bond donors and acceptors, aromatic rings, and hydrophobic centroids. In particular, the amino acids of the MCT4 model in this study that were reported to participate in the transport, plus the nearby supporting amino acids were evaluated for their potential to be exploited as unique ones, thus constituting possible pharmacological targets. Seven important features were identified in the core of the transport channel, as shown in Figure 4, namely one projected ring group (F1), five projected hydrogen acceptor sites (F2–6), and one group that represents both H-acceptor and anionic or H-donor atoms (F7). Of these features, the H-bond acceptor sites, represented in green spheres, were found to be close to conserved amino acids across species for MCT4, such as Lys40, Arg278, Tyr332, and Gly336. Arg278 in particular showed high prominence for hydrogen bonding, as it was found to be in a favorable conformation and three H-acc features were attributed. Asp274 and Arg278 formed an ion pair that is crucial for the alternate conformational changes of the transporter upon substrate translocation. One projected ring group was attributed to Tyr332 that is important for substrate recognition and translocation and implies a potent pi-interaction. Tyr332 also exhibited a high potency for hydrogen bonding while in close proximity with the remaining pharmacophore features. This residue is well conserved, and its positioning in the three-dimensional model of the transport channel of MCT4 was found to be optimal. The pharmacophore model was assessed by its potential to accommodate BNG, the co-crystallized ligand of the template structure of GLUT1 (PDB: 4PYP), and five out of seven features were satisfied.

To further validate the designed pharmacophore, we performed docking simulations of the aforementioned molecules that have inhibitory effects against MCT isoforms. In particular, syrosingopine, AR-C155858, CHC, phloretin, quercetin, AZD3965, and 7ACC2 were docked against the hMCT4 model and were evaluated for their specific ligand interactions and complex energies (Figure 5). Consistent with the experimental data, we found that syrosingopine had the lowest interaction energy and formed a hydrogen bond with Lys40 and validated the key pharmacophore features, making it most prominent both structurally and energetically (Appendix A). In Figure 6, all seven compounds are shown in their docking poses against the 3D pharmacophore.

## 3. Discussion

The phylogenetic analysis of the available sequences within the SLC16A family from mammals shows a distinct relation between the 14 isoforms. MCT1-4 are closely related and are the most abundant isoforms in mammals. More distant relations can be noted within isoforms MCT6, MCT7, MCT11–13 and MCT5, MCT14, and MCT9. MCT8 and MCT10 isoforms, which are also characterized for their transport of thyroid hormones and aromatic amino acids, respectively, belong to the same branch but have the most distant evolutionary relation with the rest family members. The phylogenetic analysis of the available full-length MCT4 proteins identified members in the kingdom Animalia and kingdom Fungi major eukaryotic taxonomic divisions. The multiple sequence alignment shows a clear distinction between fungi and insects in the length of the cytosolic loop between the N- and C-terminal domains. The extended loop length is also evident in isoforms MCT5, MCT7, MCT9, and MCT14 in mammals. However, there is no implication for a differential functional role depending on the length and thus flexibility of the interdomain loop. 

The implemented homology modelling pipeline was based on the structural homology of the MFS superfamily members and two distinctive experimentally resolved structures: the human glucose transporter GLUT1, which facilitates the translocation of glucose across the membrane and has a prominent role in cancer metabolism, and a bacterial homolog of MCT4, the *Syntrophobacter fumaroxidans* MPOB. SfMOB has the highest sequence similarity amongst the crystal structures that were identified (24.1%) for an outward–open conformation of the transporter. Even though hGLUT1 has significantly lower sequence similarity (20.9% sequence identity and 35.8% sequence similarity), it is the closest related mammal protein to hMCT4 with an experimentally determined 3D structure. Analyzing the sequence of hGLUT1, it can be noted that a higher conservation density is found at the N-terminal six-half domain. Specifically, more conserved residues are located in the membrane spanning regions of helices 1–3. This can be interpreted based on the dominant functional role of the N-terminal half in the transporter’s translocation and insertion to the membrane. Moreover, this selective conservation can be supported by the hypothesis that the two six-helix domains resulting from a gene duplication with ancient roots exert a pseudo-symmetry, and the C-terminal domain evolved for substrate specificity. Located at this region, Motif A of the MFS superfamily serves as a structural constraint for homology modelling. On the other hand, SfMOB shows a constant conservation density across the protein sequence length, and although the MCT motifs are not identified in their full length, there are adequately conserved residues within. A notably high conservation in TM helix 3 was recognized in both template structures and is possibly related to the proposed interaction of TM3 in MFS members to their chaperone (CD147-basigin for MCT4). Comparatively, TM1–3 of hGLUT1 show a finer local alignment against hMCT4 and were selected as partial templates, whilst TM4–12 were designed using SfMOB as reference. The final model was further refined based on the recently published structure of hMCT1.

The resulting 3D model of hMCT4 was energetically minimized and subjected to molecular dynamics simulations in a lipid bilayer and achieved an overall stable conformation. The model retains all characteristic features that are known for the MCT4 conformation in terms of orientation of transmembrane helices and side chain orientation for amino acids involved in proton and substrate binding. In the outward–open conformation, MCT4 forms a positively charged region in the transmembrane channel by two basic residues, Lys40 and Arg278, which are responsible for initial lactate transportation. Lys40 is protonated upon lactate binding, inducing a structural rearrangement of the N- and C-terminal domains to finally form the inward–open state. During transition, the bound lactic acid is transferred to the ion pair formed by Asp274 and Arg278, and Lys40 is deprotonated. This facilitates the release of lactic acid in the cytoplasmic medium, and the overall alternating access upon protonation and substrate binding, according to the rocker switch model (Figure 7). Transport assays and site-directed mutagenesis experiments have shown that the ion pair D274/R278 is crucial for the transport mechanism and the conformational state of the transporter. Tyr332 and Gly336, which are also involved in substrate specificity, are exposed to the channel. Compared to the recently published structure of hMCT1 in an outward–open conformation, the model exerts all known structural features with an RMSD of 0.936 Å [25]. Overall, key residues that have been identified to participate in the proton-mediated substrate translocation are exposed to the solvent, allowing for efficient pharmacophore modelling and screening for potent inhibitors of the transport channel (Appendix A). 

A simplified, reduced pharmacophore model is described herein, where the minimal set of features that need to be met, in order for a potential inhibitor compound to be in silico annotated as potentially active, contain four projected H-bond acceptor sites, two projected ring groups, and a site of either an H-acceptor group or an anionic or H-donor group. This seven-site receptor-based pharmacophore model reflects adequately the interaction area in proximity with conserved amino acids, such as Lys40, Arg278, Tyr332, and Gly336, through the transmembrane canal, for potent interaction. The concluding pharmacophore model was tested against the hGLUT1 co-crystallized ligand BNG based on the similarity of exposed residues and showed an adequate performance in the binding mode. For further validation, the model was also tested against seven compounds reported in the bibliography to have minor or major inhibitory effects against MCT isoforms, and the simulations showed that syrosingopine, a potent inhibitor against both MCT1 and MCT4, outperformed in terms of binding, ligand interactions, and pharmacophoric features the other six compounds that are not validated MCT4-specific inhibitors.

A major concern of using MCT blockers for the treatment of cancers is their probable brain toxicity, since MCTs control the delivery of lactate produced by astrocytes, to neurons, where it is used as an oxidative fuel. Cognitive defects, epilepsy, and neuronal degeneration are the consequences of chronic MCT dysfunction [10]. However, the tissue and substrate specificity of various MCTs, as well as the specificity and transient use of their blockers during cancer treatment, along with dose-dependent treatment and/or combinatory therapeutics strategies, will overcome severe toxicity problems [26]. In vitro assays on the reference compounds have shown that they exert a rather safe profile, and approved drugs or compounds under clinical trials show promising results for anti-cancer treatment (Appendix A). Syrosingopine, for instance, a dual blocker of MCT1 and MCT4, used for treatment of hypertension several decades ago, is well tolerable. Depression, lethargy or fatigue, and nasal congestion with acceptable severity were observed in some patients [27]. Benjamin et al. showed that dual inhibition of MCT1 and MCT4 with syrosingopine, thus inhibiting lactate transport, is not cytotoxic itself but only when oxidative phosphorylation is also inhibited (30540938). MCT1 is expressed ubiquitously, while MCT4 is expressed in several tissues including astrocytes [28]. The plasticity of cancer cell metabolism raises additional difficulties for an effective elimination of cancer cells. The MCT1 inhibitor AZD3965, combined with an inhibitor of oxidative phosphorylation, induced significant lymphoma cell death in vitro and reduced disease burden in vivo [29], and MCT2 is the predominant MCT in neurons [30]. MCT2 is the predominant MCT in neurons and has been shown to have neuroprotective against excitotoxicity [28]. Hence, specificity of MCT inhibitors for each isoform is very crucial for overcoming cytotoxicity in the brain, while non-selective inhibition can cause adverse side effects. However the experience with MCT4 inhibitors in clinical trials is very limited and pertains to animal studies. In a recent report nude mice received xenografts of feline oral squamous cell carcinoma, and the mice were treated with MD-1, a novel dual inhibitor of MCT1 and MCT4. A regression of the tumor but not a central nervous syndrome disturbance was reported [31]. The MCT1 inhibitor AZD3965 can be administered to patients at doses that engage the drug target, with an MTD of 20 mg od po. Dose limiting toxicities seen were primarily dose dependent, asymptomatic, and reversible changes in retinal function, which were an expected on-target effect. Investigation of the activity of AZD3965 is ongoing in tumors known to express MCT1 [32]. To conclude the issue of brain toxicity of MCT1 and MCT4 inhibitors, we should acknowledge that MCT1 inhibitors have been used in clinical research in humans, while MCT4 inhibitors have been used in preclinical research and animal studies and both constitute a promising chemotherapy for drug-resistant cancer [33]. Thus, the issue of toxicity can be overcome by the limited time of using these inhibitors and the specificity of compounds for certain MCTs. The results of the first trials in humans and animals seem to be encouraging.

The aim of this analysis was initially to interpret the unique characteristics of transported molecules and/or potential inhibitors in regard to their possible 3D conformations in the transmembrane canal of MCT4 and to provide the means required to establish a 3D-pharmacophore model for MCT4 that would enable us to screen for specific anti-MCT4 highly specific agents more accurately. This correlation was achieved by designing a tailor-made pharmacophore model based on the anatomy and physicochemical properties of MCT4. All in all, herein we attempt to assess the characteristic features of hMCT4 and the substrate translocation mechanism through the membrane in a structure-based approach. We propose a model for the outward–open conformation of hMCT4 that can be further used for molecular simulations and a pharmacophore model that can be used for the rapid and efficient screening of potential modulators of the MCT4 transporter.

## 4. Materials and Methods

### 4.1. Phylogenetic Analysis and Conserved Motif Identification

The full-length protein sequences related to the SLC16 family were extracted from the NCBI protein database [34]. In total, 48 MCT protein sequences were retrieved from mammals and 196 SLC16A3 (MCT4) related protein sequences from all available species (Appendix A). Multiple sequence alignment was performed using the MEGA software and the ClustalW algorithm [35,36]. Phylogenetic analyses were performed, and the representative phylogenetic trees were generated using Jalview software utilizing the neighbor-joining (NJ) statistical method with 100 bootstrap replicates [37].

### 4.2. Molecular Modelling and Homology Modelling

Molecular modelling, energetic calculations, and structural visualizations of all 3D structures and models were performed using the Molecular Operating Environment (MOE, version 2013.08) platform [38]. For the homology modelling purposes of this study, the amino acid sequence of MCT4 was retrieved from the Uniprot database (O15427) [39] and was used to search for homologous proteins using the DELTA-BLAST algorithm against the Protein Data Bank (PDB) [40,41]. The search algorithm reports 59 3D structures of homologous proteins that belong in the major facilitator superfamily (MFS), sharing common structural features but low sequence identity. The structure identified with the highest homology (21.77%) was the *Syntrophobacter fumaroxidans* MPOB SLC16 homolog (PDB: 6G9X) [23]. Due to the low homology level, a constrained dual template homology modelling pipeline was performed, and the suitable template structures were determined based on the sequence similarity of the 12 transmembrane regions and the source organism, using the Homology Model application of MOE. The three-dimensional model of MCT4 was based on the *Homo sapiens* glucose transporter GLUT1 (PDB: 4PYP) [24] that is identified as the human structure that shows higher similarity on transmembrane helices 1–3 and the MCT4 bacterial homologue (PDB: 6G9X), for helices 4–12, and refined based on the recently published structure of hMCT1 (PDB: 7CKR) [42]. The homology modelling was carried out under Amber10 forcefield, and a generalized Born/volume integral implicit solvent model [43]. The candidate structure with the lowest RMSD deviation from the respective template backbone was selected and superposed to the reference structures using the protein alignment tool in MOE. The resulting model was evaluated based on the sequence motif identification of the MFS members and SLC members. The model was energetically minimized and subjected to molecular dynamics simulations. Finally, the 3D geometry of the models in terms of their phi/psi angles, planarity, and rotamer strain energy profiles were evaluated using the MOE suite.

### 4.3. Energy Minimization and Molecular Dynamics Simulations

Energy minimizations were used to remove any residual geometrical strain in each molecular system, using the Charmm27 forcefield as it is implemented in MOE. The model was subjected to further calculations to evaluate the 3D geometry of the model in terms of their Ramachandran plots, omega torsion profiles, phi/psi angles, planarity, C-beta torsion angles, and rotamer strain energy profiles. The hMCT4 model was then embedded in a DPPC bilayer [44], eliminating lipid molecules within 4.5 Å of the protein, and the system was solvated with water molecules and neutralized with counter-ions as required. The complete system was minimized and then subjected to molecular dynamics (MD) simulations in a periodic environment, using the NVT ensemble. The system was simulated at 300 K using the Nosé–Poincaré–Andersen method for 100 ns.

### 4.4. Pharmacophore Elucidation

The pharmacophore model against hMCT4 model was generated using the Pharmacophore tool in MOE. The complexes of hGLUT1 with b-nonylglucoside and bacterial MCT4 homolog with (2S)-2-hydroxypropanoic acid (6HCL) were used as references to initialize the pharmacophore query. A consensus pharmacophore query for all the exposed atoms of the transport canal was then calculated with a 2 Å tolerance and resubmitted to evaluation. Docking simulations were performed using the docking module of MOE. The structures of chemical compounds were downloaded from the PubChem database [45] and were rigidly docked after minimization. Each complex was further minimized and evaluated for the resulting ligand interactions, and their energetic components were calculated. 

## Figures and Tables

**Figure 1 ijms-22-02918-f001:**
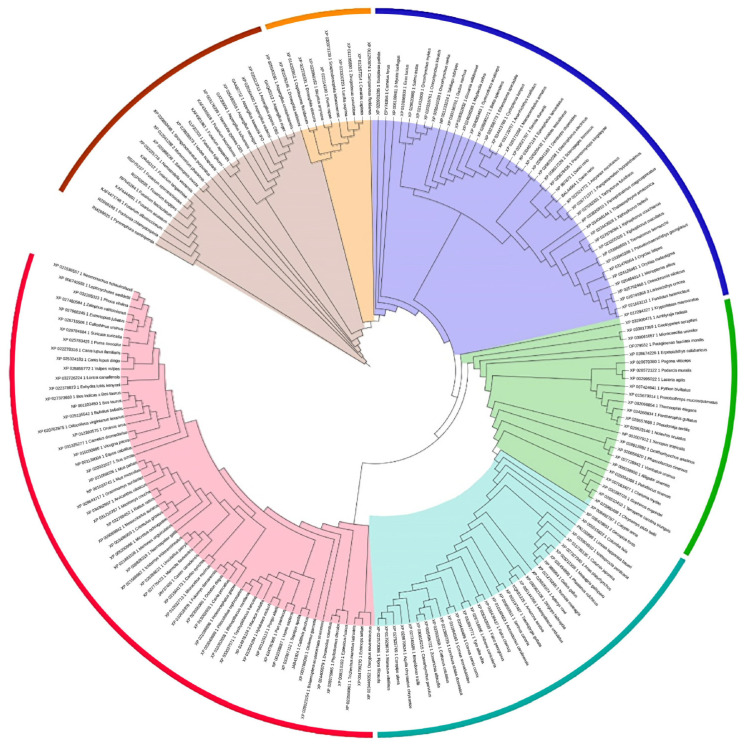
Phylogenetic tree of *SLC16A3* transporter (MCT4) across species (color coding: brown for fungi; orange for insects; dark blue for fish; green for amphibians, turtles, and lizards; light blue for birds; and red for mammals).

**Figure 2 ijms-22-02918-f002:**
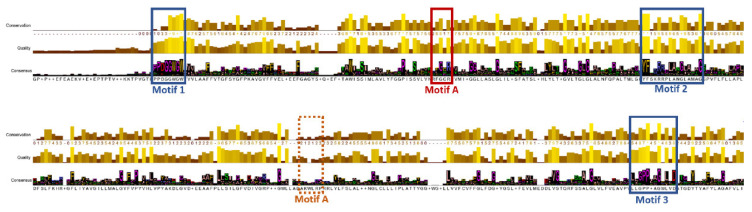
Sequence alignment of the *SLC16A* family in mammals. Depicted in blue is the MFS motif identification, in red the *SLC16A* Motif A, and in orange the expected location of the repeated Motif A in loop L8–9.

**Figure 3 ijms-22-02918-f003:**
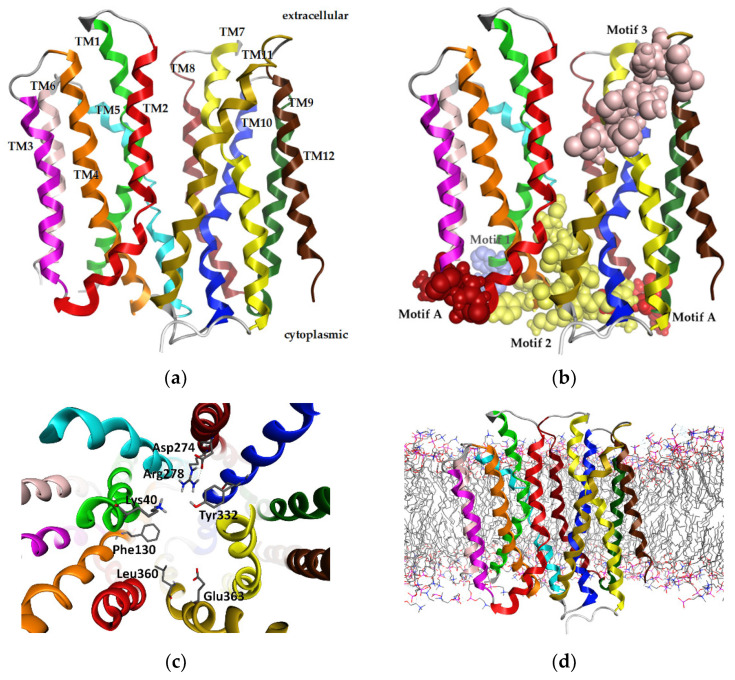
The overall structure of the human MCT4 model. (**a**) Ribbon representation of hMCT4 3D model colored by transmembrane helix. (**b**) Sphere representation of the identified motifs of hMCT4. (**c**) Stick representation of the key residues in the interacting area of MCT4 transport mechanism, including key residues for proton interaction and residues involved in the substrate translocation. (**d**) The model embedded in the membrane.

**Figure 4 ijms-22-02918-f004:**
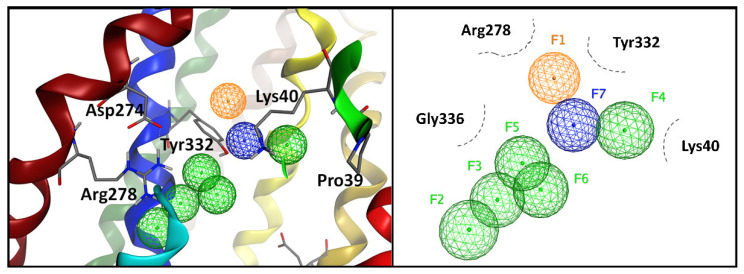
The pharmacophore model of MCT4 in 3D (**left**) and 2D (**right**). The pharmacophore model consists of seven points that can be used to describe the properties of the most potent and promising compounds in this study. Green grid spheres represent projected H-acc groups, orange grid spheres represent a projected ring group, and blue grid spheres represent a projected group of both H-donor and H-acceptor.

**Figure 5 ijms-22-02918-f005:**
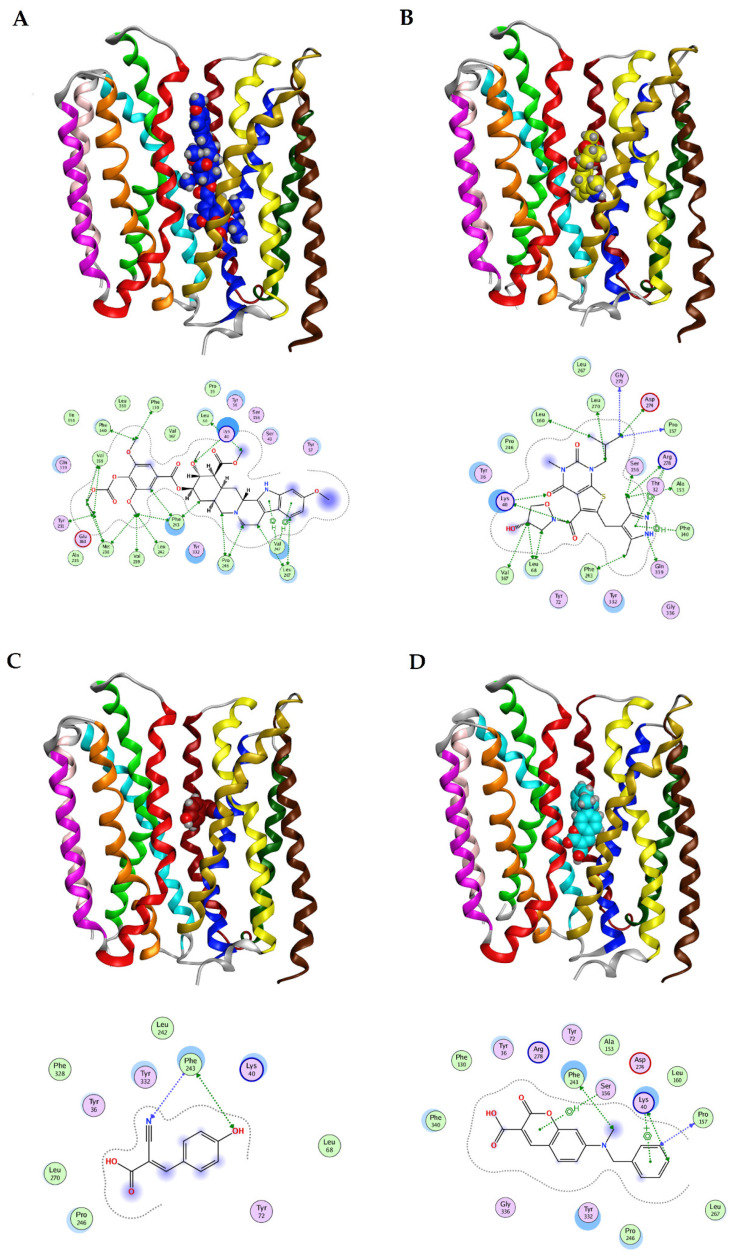
Docking positions and ligand interactions for the seven compounds simulated in sphere representation. (**A**) Syrosingopine in blue, (**B**) AR-C155858 in yellow, (**C**) CHC in red, (**D**) 7ACC2 in light blue, (**E**) quercetin in pink, (**F**) phloretin in magenta, and (**G**) AZD3965 in green.

**Figure 6 ijms-22-02918-f006:**
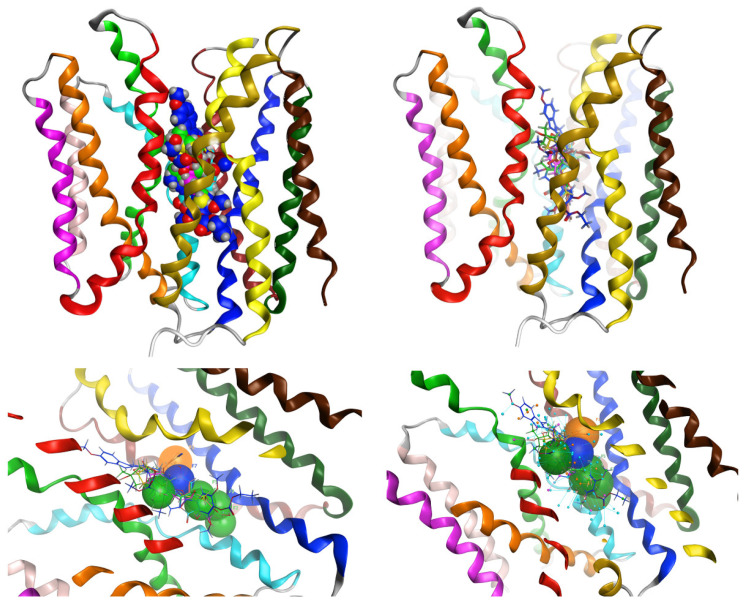
The seven compounds in their docking positions in sphere representation (**top left**) and stick representation (**top right**) and their conformations against the pharmacophoric features for the hMCT4 model (**bottom**).

**Figure 7 ijms-22-02918-f007:**
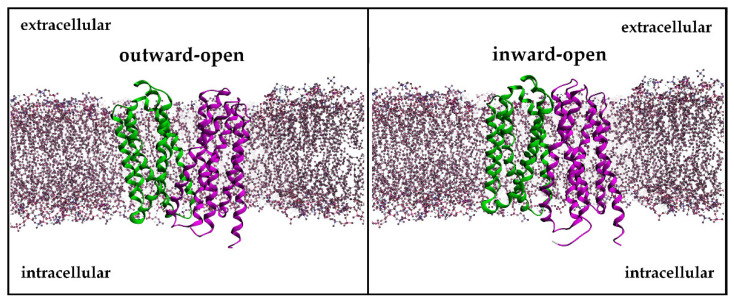
The 3D representation of the outward–open (**left**) and inward–open (**right**) conformation of the transmembrane transporters embedded in the membrane. The structures represent the hMCT4 model in an outward and inward state.

## Data Availability

Publicly available datasets were analyzed in this study. This data can be found here: https://www.ncbi.nlm.nih.gov/protein/, https://www.uniprot.org/uniprot/O15427, https://www.rcsb.org/structure/6G9X, https://www.rcsb.org/structure/4pyp, https://www.rcsb.org/structure/7CKR (accessed on 1 February 2021).

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
