# Peer review of "A Holistic Evolutionary and 3D Pharmacophore Modelling Study Provides Insights into the Metabolism, Function, and Substrate Selectivity of the Human Monocarboxylate Transporter 4 (hMCT4)"

_ijms, 2021, doi:10.3390/ijms22062918_

Round 1

Reviewer 1 Report

In this article, Papakonstantinou and colleagues aimed to elucidate the structural features of MCT4 for specified substrate recognition using in silico molecular dynamics simulations as a therapeutic approach towards the identification of any potent inhibitors of MCT4. The authors highlight the point that one promising therapeutic strategy in anti-cancer drug design is the selective inhibition of MCT4 for the glycolytic suppression of solid tumours. There have been a number of studies carried out that have assessed candidates for dual inhibition of both MCT1 and MCT4 but as the authors rightly point out, no selective inhibitor with high-affinity for MCT4 has been identified as yet.This is an important issue to address for very many people and so it is a priority for a number of research groups.

Under the experimental conditions used in these analyses, the authors initial aim was to interpret the unique characteristics of transported molecules and / or potential inhibitors with regards to their possible 3D conformations in the transmembrane canal of MCT4 and to provide the means required to establish a 3D-pharmacophore model that would enable them to more accurately screen for relevant anti-MCT4 high specific agents. The authors ultimately achieved this correlation by designing a tailor-made pharmacophore model based on the anatomy and physicochemical properties of MCT4.

The article does what the title suggests and is an interesting read.

Main points and comments:

  1. The authors have carried out a logical set of analyses and have explained their rationale throughout the article. The Materials and Methods all seem appropriate for this style of analysis and the resultant models are clearly presented and annotated. This is basically a nicely presented article with some interesting ideas and suggestions regarding anti-cancer therapies and strategies. The ribbon and sphere representations are all easy to follow and decipher.

  1. However, there is a little bit of confusion regarding some of the referencing of Figures and Tables in the text. This needs to be corrected in order to make the article easier to follow. At present it is a bit of a jumble. Can the authors ensure that all Figures and Tables (including the Supplementary data), are quoted correctly in the text throughout the article please?

Line 167. I think the authors mean Figure 3b (rather than Figure 4b as shown).

Lines 326 -327. I think this should read Table S2 and not Table S1? Can the authors please check this and correct as required?

  1. The authors have utilised the basis of scientific and systematic study of the pharmacophore model of MCT4 in both 3-D and 2-D and have identified seven important features in the core of the transport channel (as demonstrated clearly in Figure 4). The concept of computer-aided drug design has been around for a while and the scientific use of 3-D pharmacophore modelling does provide an interesting and potentially very successful method for elucidating essential molecular interactions between any ligand and target in the form of a comprehensive spatial arrangement. The use of pharmacophore modelling does enable virtual screening of structural characteristics of MCT4 in order to determine the identification of specific inhibitors that can then be used as novel anti-cancer compounds. I think these are very neat in-silico experiments and they generate vast amounts of information in a reasonably short space of time.

The combination of 3D pharmacophore modelling with molecular dynamics simulations have enabled these approaches to consider macromolecule–ligand interactions as dynamic and have therefore demonstrated a physiologically relevant interaction pattern in the output data.

  1. Can the authors comment on the possibility of brain toxicity during the use of MCT blockers in cancer treatment? How big an issue is this?

  1. Can the authors make any further quantitative predictions from their models?

  1. The Supplementary data files are very useful and informative. However, I do question how much of Figure S2 is actually going to be seen by the reader as it is so small. I realise we all do this at some stage and assume that it will be OK. I know you can obtain a broad indication of the sequence alignment without seeing all the minor detail but it would be nice to be able to read the Figure fully at the publishable size.

  1. Can the authors comment on the mechanistic outcome of the protein functionality?

  1. There are a few small typographical errors here and there that need correcting or the use of an incorrect word in some sentences (such as lines 29, 30, 53, 71, 78, 204, 215, 301, 400).

  1. I would suggest using the phrase “top right” and “top left” for Figure 6 (rather than “up right” and “up left” as in lines 256 and 257).

The authors have basically proposed a model for the outward-open conformation of hMCT4 that can now be used further for molecular simulations and a pharmacophore model that can be used for the rapid and efficient screening of potential modulators of the MCT4 transporter.  The authors have generated some elegant data and have presented their results in a clever way to highlight all the analyses.

This is an interesting article that just requires a little bit of proof-reading in order to iron out one or two inconsistencies, along with careful numbering of the Figures and Tables throughout.

Author Response

Point-by-point response to the Reviewers’ comments

REVIEWER’S 1 EVALUATION

Comments and Suggestions for Authors

In this article, Papakonstantinou and colleagues aimed to elucidate the structural features of MCT4 for specified substrate recognition using in silico molecular dynamics simulations as a therapeutic approach towards the identification of any potent inhibitors of MCT4. The authors highlight the point that one promising therapeutic strategy in anti-cancer drug design is the selective inhibition of MCT4 for the glycolytic suppression of solid tumours. There have been a number of studies carried out that have assessed candidates for dual inhibition of both MCT1 and MCT4 but as the authors rightly point out, no selective inhibitor with high-affinity for MCT4 has been identified as yet.This is an important issue to address for very many people and so it is a priority for a number of research groups.

Under the experimental conditions used in these analyses, the authors initial aim was to interpret the unique characteristics of transported molecules and / or potential inhibitors with regards to their possible 3D conformations in the transmembrane canal of MCT4 and to provide the means required to establish a 3D-pharmacophore model that would enable them to more accurately screen for relevant anti-MCT4 high specific agents. The authors ultimately achieved this correlation by designing a tailor-made pharmacophore model based on the anatomy and physicochemical properties of MCT4.

The article does what the title suggests and is an interesting read.

- Thank you very much for your respectful comments.

Main points and comments:

  1. The authors have carried out a logical set of analyses and have explained their rationale throughout the article. The Materials and Methods all seem appropriate for this style of analysis and the resultant models are clearly presented and annotated. This is basically a nicely presented article with some interesting ideas and suggestions regarding anti-cancer therapies and strategies. The ribbon and sphere representations are all easy to follow and decipher.

- Thank you very much for your respectful comments.

  1. However, there is a little bit of confusion regarding some of the referencing of Figures and Tables in the text. This needs to be corrected in order to make the article easier to follow. At present it is a bit of a jumble. Can the authors ensure that all Figures and Tables (including the Supplementary data), are quoted correctly in the text throughout the article please?

- Thank you very much for your comments. All Figures and Tables have been quoted correctly as advised. In detail:

Lines 144,146 The correct figures are referenced.

Line 386 The correct Table S2 is referenced.

Line 167. I think the authors mean Figure 3b (rather than Figure 4b as shown).

- Thank you very much for your comments. It has been corrected as advised.

Lines 326 -327. I think this should read Table S2 and not Table S1? Can the authors please check this and correct as required?

- Thank you very much for your comments. It has been corrected as advised.

  1. The authors have utilised the basis of scientific and systematic study of the pharmacophore model of MCT4 in both 3-D and 2-D and have identified seven important features in the core of the transport channel (as demonstrated clearly in Figure 4). The concept of computer-aided drug design has been around for a while and the scientific use of 3-D pharmacophore modelling does provide an interesting and potentially very successful method for elucidating essential molecular interactions between any ligand and target in the form of a comprehensive spatial arrangement. The use of pharmacophore modelling does enable virtual screening of structural characteristics of MCT4 in order to determine the identification of specific inhibitors that can then be used as novel anti-cancer compounds. I think these are very neat in-silico experiments and they generate vast amounts of information in a reasonably short space of time.

The combination of 3D pharmacophore modelling with molecular dynamics simulations have enabled these approaches to consider macromolecule–ligand interactions as dynamic and have therefore demonstrated a physiologically relevant interaction pattern in the output data.

- Thank you very much for your respectful comments.

  1. Can the authors comment on the possibility of brain toxicity during the use of MCT blockers in cancer treatment? How big an issue is this?

- Thank you very much for your consideration. An additional input is included in Lines 382-414 of the current revised manuscript that mentions brain toxicity issues for MCT blockers. Indeed, toxicity is a major concern in rational drug design, and especially when targeting MCT isoforms expressed on the brain (MCT1/2/4), overcoming this issue is quite challenging. MCT isoforms play a major role in glycolytic cells metabolism and are known to be overexpressed in pathological states, such as cancer or ischemia, as well as in normal conditions, like exercise, and deficient function of lactate transport in the brain can lead to neuropathogenesis. However the experience with MCT4 inhibitors in clinical trials is very limited and pertains mainly to animal studies. Thus, the issue of toxicity can be overcome by the limited time of using these inhibitors and the specificity of compounds for certain MCTs. The results of the first trials in humans and animals seem to be encouraging.

  1. Can the authors make any further quantitative predictions from their models?

- Thank you very much for your consideration. Indeed, further in silico experiments have been conducted by the authors and a sample of the results from pharmacophore-based high throughput screening are included in Figure S4 of the Supplementary Material.

  1. The Supplementary data files are very useful and informative. However, I do question how much of Figure S2 is actually going to be seen by the reader as it is so small. I realise we all do this at some stage and assume that it will be OK. I know you can obtain a broad indication of the sequence alignment without seeing all the minor detail but it would be nice to be able to read the Figure fully at the publishable size.

- Thank you very much for your comments, Figure S2 has been changed accordingly. The frame has been enlarged and the additional information of hMCT1 has been included in a way that is easily readable in detail.

  1. Can the authors comment on the mechanistic outcome of the protein functionality?

- Thank you very much for your consideration. An additional input is included in Lines 333-343 of the current revised manuscript that describes the mechanistic basis of MCT4 transport accompanied with an additional figure (Figure 7). The protein follows a rocker-switch mechanism for substrate translocation alternating its conformation from an outward open to an inward open state. These changes are mediated by specific binding of key residues mentioned in the manuscript.

  1. There are a few small typographical errors here and there that need correcting or the use of an incorrect word in some sentences (such as lines 29, 30, 53, 71, 78, 204, 215, 301, 400).

- Thank you very much for your comments, the authors have made a careful proof-reading and believe that all errors have been corrected.

  1. I would suggest using the phrase “top right” and “top left” for Figure 6 (rather than “up right” and “up left” as in lines 256 and 257).

- Thank you very much for your comments, it has been revised as requested.

The authors have basically proposed a model for the outward-open conformation of hMCT4 that can now be used further for molecular simulations and a pharmacophore model that can be used for the rapid and efficient screening of potential modulators of the MCT4 transporter.  The authors have generated some elegant data and have presented their results in a clever way to highlight all the analyses.

This is an interesting article that just requires a little bit of proof-reading in order to iron out one or two inconsistencies, along with careful numbering of the Figures and Tables throughout.

- Thank you very much for your respectful comments.

Reviewer 2 Report

The Authors provided in silico studies of the MCT system. The topic is suitable for the IJMS scope. The manuscript is quite well organized, but major revision is needed due to figures quality. Please highlight the most important achievements. Some detailed comments are listed below. In the abstract should be indicated that the experiment based on the modeling.
l. 53 CO2 - please change it
l. 60 "may be" or are?
explain AR-C155858 and  AZD3965
l.133 explain NJ
Fig. 1 change it or give in supporting materials detailed description
Fig. 3 enlarge the fonts
Figs 4-6 should be changed because it is impossible to recognize the letters and numbers, and structures. I.e fig 5 can be enlarged to a full page.
In the results and discussion section, the most important achievements (results) should be somehow highlighted.
l.343 please express which model - describe it more

Author Response

REVIEWER’S 2 EVALUATION

Comments and Suggestions for Authors

The Authors provided in silico studies of the MCT system. The topic is suitable for the IJMS scope. The manuscript is quite well organized, but major revision is needed due to figures quality. Please highlight the most important achievements. Some detailed comments are listed below. In the abstract should be indicated that the experiment based on the modeling.

- Thank you very much for your respectful comments. The abstract in the revised manuscript includes the molecular modelling strategy to clarify the computational approach that was followed.

  1. 53 CO2 - please change it

- Line 53(now 57) of original manuscript: CO2 has been changed to carbon dioxide.

  1. 60 "may be" or are?

- Line 60 (now 63) of original manuscript: “…may be the fundamental…” has been changed to “…are a fundamental...”

explain AR-C155858 and  AZD3965

- Line 113 (now 120) of original manuscript: AR-C155858 is now explained as a cell-permeable thieno[2,3-d]pyrimidinedione compound.

- Line 115 (now 122) of original manuscript: AZD3965, is now explained as a pyrrole pyrimidine derivative.

l.133 explain NJ

- Line 133 (now 143) of original manuscript: NJ has been changed to Neighbor-Joining (NJ).

Fig. 1 change it or give in supporting materials detailed description

- Thank you very much for your consideration. Additional phylogenetic tree with representative species have been added in the Supplementary Material under Figure S1.

Fig. 3 enlarge the fonts

- Figure 3 has changed according to the comments of the respected Reviewer, it is more readable, and the fonts are enlarged.

Figs 4-6 should be changed because it is impossible to recognize the letters and numbers, and structures. I.e fig 5 can be enlarged to a full page.

- Figures 4-6 have been changed according to the comments of the respected Reviewer, they are more readable, and the fonts are enlarged.

In the results and discussion section, the most important achievements (results) should be somehow highlighted.

- Thank you very much for your consideration. An additional input is included in Lines 334-347, Lines 379-381 and  Lines 394-426 in the Discussion session of the current revised manuscript highlighting the most important results and the importance of MCT inhibition as an anti-cancer strategy.

l.343 please express which model - describe it more

An analysis of the hMCT4 pharmacophore model consisting of 7 property associated locations in the hMCT4 binding site is analysed in the Results section (Lines 217-237) and pictured in Figure 4.

- Thank you very much for your very constructive comments and help. All suggestions have been considered.

Round 2

Reviewer 2 Report

The Authors improved the text due to reviewers' comments. One general remark: please put in supporting data detailed description which was presented in fig. 1 - I see some letters (names of species?) but it is impossible to recognize it. You could make some tables in supporting materials. 

Just one editorial remark: please verify the quality of the figures before final and check are there in the proper place and proper size.

Author Response

Point-by-point response to the Reviewers’ comments

REVIEWER’S 2 SECOND ROUND EVALUATION

Comments and Suggestions for Authors

The Authors improved the text due to reviewers' comments. One general remark: please put in supporting data detailed description which was presented in fig. 1 - I see some letters (names of species?) but it is impossible to recognize it. You could make some tables in supporting materials. 

Just one editorial remark: please verify the quality of the figures before final and check are there in the proper place and proper size.

A high resolution Figure 1 has been included as Figure S1 in the supplementary material together with Table S3 that contains all the sequence information of the phylogenetic tree presented. Figure 2 has been replaced by a higher resolution image making the sequence below readable on magnification.